# Chemoselective synthesis and analysis of naturally occurring phosphorylated cysteine peptides

Jordi Bertran-Vicente[1], Martin Penkert[1,2], Olaia Nieto-Garcia[1,2], Jean-Marc Jeckelmann[3], Peter Schmieder[1], Eberhard Krause[1] & Christian P. R. Hackenberger[1,2]

In contrast to protein O-phosphorylation, studying the function of the less frequent N- and S-phosphorylation events have lagged behind because they have chemical features that prevent their manipulation through standard synthetic and analytical methods. Here we report on the development of a chemoselective synthetic method to phosphorylate Cys side-chains in unprotected peptides. This approach makes use of a reaction between nucleophilic phosphites and electrophilic disulfides accessible by standard methods. We achieve the stereochemically defined phosphorylation of a Cys residue and verify the modification using electron-transfer higher-energy dissociation (EThcD) mass spectrometry. To demonstrate the use of the approach in resolving biological questions, we identify an endogenous Cys phosphorylation site in IICB$^{Glc}$, which is known to be involved in the carbohydrate uptake from the bacterial phosphotransferase system (PTS). This new chemical and analytical approach finally allows further investigating the functions and significance of Cys phosphorylation in a wide range of crucial cellular processes.

[1] Leibniz-Institut für Molekulare Pharmakologie (FMP), Robert-Rössle-Str. 10, D-13125 Berlin, Germany. [2] Humboldt-Universität zu Berlin, Institut für Chemie, Brook-Taylor-Str. 2, D-12489 Berlin, Germany. [3] Institute of Biochemistry and Molecular Medicine, University of Bern, 3012 Bern, Switzerland. Correspondence and requests for materials should be addressed to C.P.R.H. (email: hackenbe@fmp-berlin.de).

Protein phosphorylation is a major actor in the regulation of biochemical signalling pathways and a wide range of other essential cellular processes[1,2]. Extensive studies of phosphorylation, which mainly occurs on serine, threonine and tyrosine amino-acid side chains, have revealed important insights into protein functions. The phosphorylation of other amino acids, such as phospho-histidine (pHis), -arginine (pArg), -lysine (pLys) and -cysteine (pCys), is less understood and studies of these events have been hindered by technical limitations[3–7]. The main obstacle to studying N- and S-phosphorylation has been their acid-lability, which prevents their accessibility and characterization through standard chemical and analytical tools.

Important steps have been recently made towards the development of synthetic tools to allow the site-specific phosphorylation of native pLys and pArg peptides and mimetics of pHis in proteins[8–13]. These chemical tools have permitted the development of mass spectrometry (MS)-based methods or the generation of antibodies, leading to a better understanding of the biological role of these modifications[14–16]. The isolation and characterization of phosphorylated-Cys peptides, however, has been challenging due to the acid-lability of the phosphorothiolate bond, which has prevented further identification of unknown pCys sites using standard phosphoproteomic approaches[7].

pCys is known to function as an intermediate in the phosphoenolpyruvate (PEP)-dependent phosphotransferase system (PTS), in the dephosphorylation of phosphotyrosine residues by protein tyrosine phosphatases, and in bacterial signalling and regulation[17–26]. In addition to its natural biological functions, pCys has recently been employed as a phosphoserine and phosphothreonine mimetic[27–29]. In this work, Davis and co-workers[30] have developed a two-step method for installing pCys residues on a protein level; however, this protocol delivers an epimeric mixture of pCys proteins, since it relies on the reaction of dehydroalanine (Dha) with sodium thiophosphate. Although this elegant strategy provided access to pCys residues in proteins, the elimination conditions required to prepare Dha residues and the lack of stereoselectivity may impose certain limitations on the general applicability of this chemical tool in the functional analysis of phosphorylated Cys residues[8].

In this report we develop a novel chemoselective and stereochemically defined phosphorylation strategy for Cys residues (Fig. 1). As a key step in our strategy we envision to exploit the nucleophilic reactivity of P(III)-reagents (phosphites) with electrophilic disulfides. Moreover, we develop a benign MS-based proteomic approach to identify and characterize pCys sites that naturally occur in peptides. Finally, we identify and characterize an endogenous pCys peptide from the glucose-specific transporter IICB[Glc], which is known to be involved in the PTS by following a MS-based proteomic approach using electron-transfer higher-energy dissociation (EThcD) tandem MS. This work provides a novel synthetic strategy to incorporate native L-pCys residues into unprotected peptides, along with an analytical method that unambiguously reveals phosphorylation on Cys sites in native peptides.

## Results

**Synthesis of pCys peptides.** To show the feasibility of this chemical strategy, our initial experiments were carried out in reactions with naturally occurring cystines using phosphite triesters that had previously been synthesized by our laboratory (**3b-e**) (refs 9,10). Our first attempts to use oxidized glutathione (GSSG) and commercially available trimethyl phosphite in dimethylformamide (DMF) did not reveal a formation of a corresponding phosphorothiolate ester. The reactivity of GSSG towards phosphites was also tested in water (25 mM Tris-HCl at pH 8.0) using the water-soluble phosphite **3e**, but once again failed to exhibit a formation of phosphorothiolate esters.

Based on reports of previous work on the synthesis of phosphoro-thioate and -thiolate nucleotides[31–33], we decided to test an electron-deficient disulfide peptide for reactivity (Fig. 1). Thereby, we take advantage of standard peptide synthesis protocols to obtain electrophilic disulfides **2** from Cys-containing peptides **1** through the addition of 5,5′-dithiobis-2-nitrobenzoic acid (Ellman's reagent). Subsequent reaction with phosphites **3** delivers the corresponding phosphorothiolate ester peptides **4**, which yield pCys-peptides **6** after phosphodiester cleavage. The use of electron-deficient aryl disulfides such as 2,2′-dithiobis(5-nitropyridine) and Ellman's reagent have been exploited to activate Cys thiols for subsequent ligation with Cys-containing ubiquitin fragments and as precursors of Dha residues upon elimination using phosphines or 1,8-diaza-bicycloundec-7-ene (refs 34,35). To the best of our knowledge, the chemoselectivity of P(III) reagents has not previously been successfully exploited in reactions between different nucleophilic phosphite esters and electrophilic disulfides to form phosphorothiolate ester derivatives on a peptide level.

We first synthesized a small Cys-containing peptide (sequence LYRCAK) **1a** using standard solid-phase peptide synthesis (SPPS; Fig. 2a). Peptide **1a** was then reacted with Ellman's reagent following standard protocols to form peptide **2a** (Fig. 2a)[35]. With the isolated peptide **2a** in hand, we first probed the reactivity with phosphites **3** in DMF (Fig. 2a, Table 1). The reaction with tributyl phosphite (**3a**) and analysis of the reaction crude by ultra performance liquid chromatography-ultraviolet (UPLC-UV) coupled to MS showed a formation of mainly Cys peptide **1a** along with a minor amount of phosphorothiolate ester peptide **4a** (Table 1). In contrast, the base-cleavable phosphite **3b** revealed a good conversion after isolation using semi-preparative high-performance liquid chromatography (HPLC), yielding peptide **4b** in 37% isolated yield (Table 1). However, the formation of peptide **1a** was also observed.

**Figure 1 | Synthetic strategy to install pCys residues in unprotected peptides.** Site-selective addition of phosphite to an activated Ellman-disulfide to deliver a stereochemically defined pCys residue.

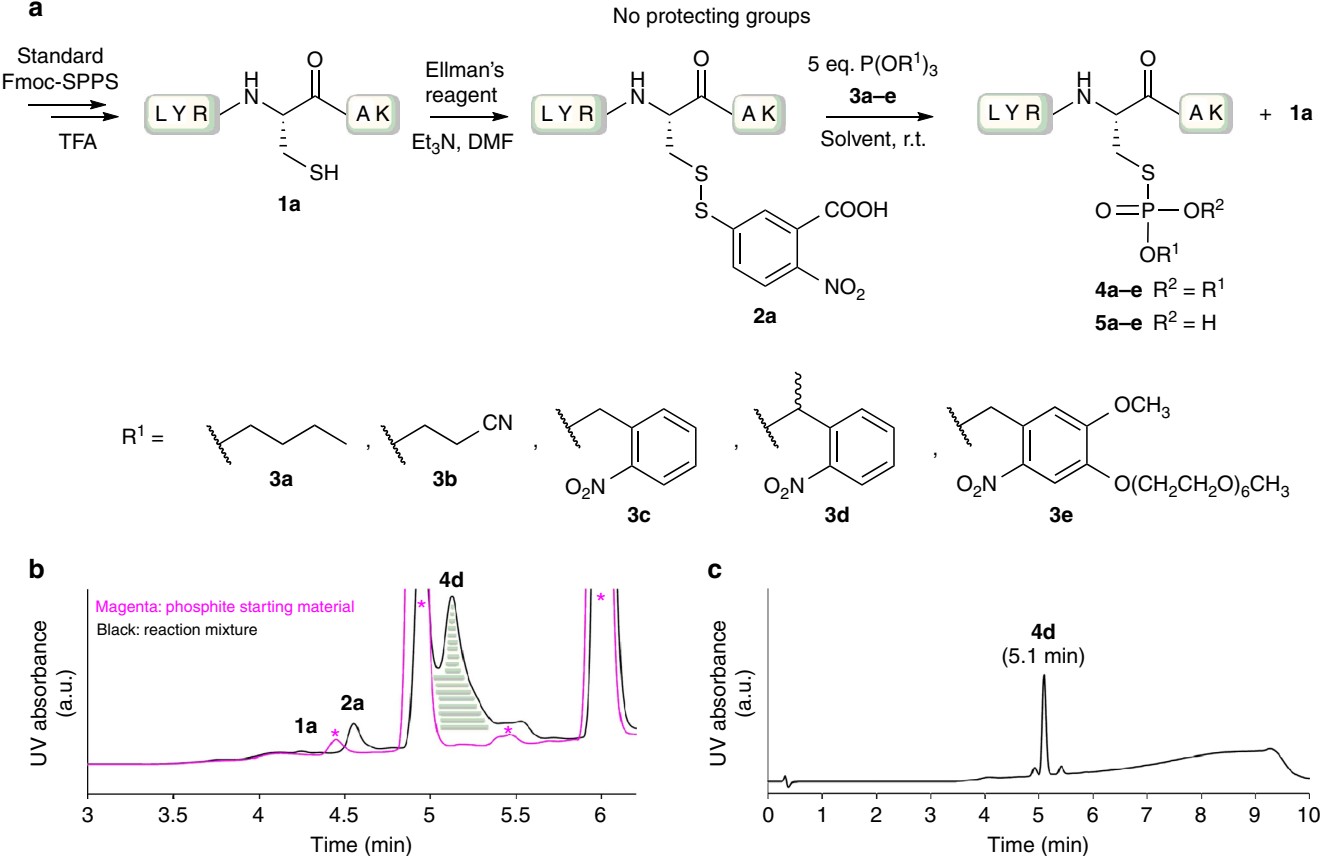

**Figure 2 | Chemoselective synthesis of phosphorothiolate esters peptides 4 and 5. (a)** Reaction of peptide **1a** with Ellman's reagent and peptide **2a** with 5 eq. phosphite esters **3a–e**. (**b**) Overlap of UPLC traces of reaction mixture for peptide **2a** and phosphite **3d** after 16 h (black) and phosphite **3d** starting material (magenta). Formation of the phosphorothiolate ester **4d** after 16 h incubation with phosphite **3d** in DMF at room temperature. Magenta asterisk shows the by-products of the phosphite **3d** decomposition, which overlap with the reaction mixture UPLC trace for peptide **2a** and phosphite **3d**. (**c**) UPLC-UV trace after purification by semi-preparative HPLC of phosphorothiolate ester **4d**. a.u., arbitrary unit; r.t., room temperature.

| Table 1 | MS-conversion and isolated yields for phosphorothiolate esters peptides 4 and 5. | | | | | | | |
|---|---|---|---|---|---|---|---|---|
| Entry | Phosphite* | Solvent† | Product | % Conversion‡ | % Yield§ | Product | % Conversion‡ | % Yield§ |
| 1 | **3a** | DMF | **4a** | 20 | — | **5a** | — | — |
| 2 | **3b** | DMF | **4b** | 54 | 37 | **5b** | — | — |
| 3 | **3c** | DMF | **4c** | 6 | — | **5c** | 32 | — |
| 4 | **3d** | DMF | **4d** | 97 | 55 | **5d** | — | — |
| 5 | **3e** | DMF | **4e** | 22 | 27 | **5e** | 35 | 34 |
| 6 | **3a** | MeCN:Tris (1:1) | **4a** | 18 | — | **5a** | — | — |
| 7 | **3b** | MeCN:Tris (1:1) | **4b** | 59 | 43 | **5b** | — | — |
| 8 | **3c** | MeCN:Tris (3:2) | **4c** | 7 | — | **5c** | 18 | — |
| 9 | **3d** | MeCN:Tris (3:2) | **4d** | 36 | 27 | **5d** | — | — |
| 10 | **3e** | Tris, pH 7.2 | **4e** | 2 | — | **5e** | 25 | — |
| 11 | **3e** | Tris, pH 8.0 | **4e** | 3 | — | **5e** | 30 | 38 |

*5 eq. of phosphite was used for each entry.
†MeCN:Tris mixtures consist on MeCN and 25 mM Tris-HCl at pH 7.2.
‡Conversion was determined by LC–MS analysis after **2a** was depleted. The sum of areas under the XIC peaks corresponding to **1a**, **2a**, oxidized **1a** and **4** and **5**, for each entry, were counted as a 100%. All experiments were performed at least in duplicate, and the average values are presented.
§Isolated yields of **4b**, **4d**, **4e** and **5e** products after semi-preparative HPLC. For synthetic procedures see Supplementary Methods.

Upon probing light-cleavable *o*-nitrobenzyl-based phosphites **3c–3e**, the benzyl-substituted phosphite **3d** showed an almost complete conversion to phosphorothiolate ester peptide **4d,** with a minor formation of hydrolysis product **1a** based on UPLC-UV-MS, and in 55% isolated yield after semi-preparative HPLC (Fig. 2a–c, Table 1). Interestingly, when using phosphites with no substituent at the benzyl position such as **3c** or **3e**, we observed formation of monoprotected phosphorothiolate ester **5** in

addition to the expected product **4** and peptide **1a** (Table 1, Supplementary Fig. 1). We reasoned that the formation of monoprotected phosphorothiolate esters (**5c**, **5e**) is due to a nucleophilic attack on the electrophilic benzylic position of the *o*-nitrobenzyl substrates by water or by the thiolate nitrobenzoic acid by-product. As expected, when using phosphite **3d** the monoprotected ester **5d** was not formed due to the decreased reactivity of the benzylic position (Fig. 2b, Table 1).

Next, we investigated the reactivity in buffered aqueous solvents. Reactions of peptide **2a** with phosphites **3a** and **3b** in a mixture of MeCN and 25 mM Tris-HCl pH 7.2 yielded conversions similar to those we had previously observed in DMF (Table 1). In the case of phosphite **3d**, we observed lower conversion to the desired product **4d**, which was isolated in 27% yield (Table 1). Interestingly, MS-readouts showed an increase in the formation of peptide **1a** and its corresponding oxidized disulfide product (Supplementary Figs 2 and 3). Using the water-soluble phosphite **3e**, the monoprotected phosphorothiolate ester **5e** was obtained in good conversions, together with formation of significant amounts of hydrolysed peptide **1a** (Table 1, Supplementary Fig. 4). In general, we reasoned that formation of peptide **1a** might occur from the first intermediate during the reaction between phosphites and electrophilic disulfides, that is, the triester thiophosphonium cation. At this stage the P-S versus P-O bond hydrolysis seems to depend heavily on the nucleophilic substances in the reaction media and also on the nature of the phosphorothiolate ester substituents that are used.

We used a small peptide sequence (YCA) with an L-Cys (**1b**) or a D-Cys (**1c**) to demonstrate that our synthetic strategy is epimerization-free. The disulfides **2b** and **2c** reacted with phosphite **3b** to produce unique phosphorothiolate ester peptides, **4h** and **4i**, respectively (Supplementary Figs 5 and 6). Analysing the reaction crudes for both phosphorylation reactions through UPLC-UV and -MS revealed that **4h** and **4i** have different retention times, which showed that our synthetic pathway does not involve a Dha intermediate (Supplementary Fig. 7). We obtained analogous results using phosphite **3a** and **3d** (Supplementary Figs 8–13).

We then applied ultraviolet-irradiation or alkaline deprotection conditions to peptides **4b**, **4d** and **4e** to facilitate the formation of native phosphorylated Cys peptides **6**. Previously, o-nitrobenzyl esters-type caged groups have been used to access pTyr analogues, native pLys peptides, or phosphomonoesters through mild light-induced photolysis[9,36–38]. The base-labile cyanoethyl group is commonly used as protecting group in phosphorimidites in a similar way for the solid-phase synthesis of oligonucleotides and we have recently used it as a precursor of pLys peptides[10,39]. The photodeprotection of peptide **4d** and **4e** upon exposure to 295 nm light for 5 and 15 min, respectively, delivered phosphorylated Cys peptide **6a**, with minor formation of the hydrolysed P-S bond by-product (Fig. 3a,b, Supplementary Fig. 14). In contrast, incubating peptide **4b** with 250 mM NaOH delivered a significant amount of Dha-containing peptide in addition to the phosphorylated Cys peptide **6a**, convincing us to abandon cyanoethyl-derivatives as protective groups for the delivery of phosphorylated Cys-containing peptides (Supplementary Fig. 15).

After we had successfully synthesized pCys peptide **6a**, we probed its stability at different pH values, using UPLC-UV and -MS to follow its rate of decay. Acidic pH values decreased the stability of the phosphorothiolate bond in pCys peptide **6a**, as expected, with a half-life of 3 h in either 0.1% formic acid (FA, pH 2.9) or 0.1% trifluoroacetic acid (TFA, pH 2.0) (Supplementary Fig. 16). In contrast, peptide **6a** showed sufficient stability with minor hydrolysis both at pH 7.4 and 8.4 over a period of 24 h (Supplementary Fig. 16). In order to avoid significant hydrolysis, pCys peptides should be purified by using a MeCN/water gradient in alkaline aqueous buffer (pH 8.4) as a mobile phase. Peptide **6a** was isolated in 65% yield and with a purity criteria $\geq$ 90% based on UPLC-UV (Supplementary Fig. 17).

**Analytical characterization of pCys peptides.** Next, we characterized the synthesized pCys peptide sequence **6a** by NMR and

MS to confirm the site-specific modification of the peptide and to rule out a phosphate-transfer or migration processes during their synthesis. Using [31]P-NMR, we observed a major signal at 12.02 p.p.m. (Fig. 3c), which is characteristic according to the literature[22,24]. Additionally, a minor peak at $-2.63$ p.p.m. was found which was attributed to inorganic phosphate by addition of external sodium phosphate. To demonstrate that the [31]P signal was attached to a Cys residue, an [1]H-[31]P HMBC NMR experiment was performed. The phosphorous signal peak at 12.02 p.p.m. showed coupling to the α-methylene hydrogen atoms in the pCys side-chain at 3.13 p.p.m., confirming that the phosphate was attached to a Cys side chain (Supplementary Fig. 18). However, if peptide sequences in biological samples had additional Cys residues, it would be difficult or impossible with NMR to determine which Cys side chains had been phosphorylated.

We, therefore, turned to MS to unambiguously localize the phosphorylation site in **6a**. In addition, we intended to implement a tandem MS (MS/MS) method, which would also facilitate the assignment of Cys phosphorylation sites in proteomic analysis of samples from biological origin[40]. MS/MS has become the method of choice for analysis of post-translational modifications in peptides; when coupled with nanoliquid chromatography (nLC), the method provides high-sensitivity and can be implemented in a high-throughput manner[41]. While collision-induced dissociation (CID) is the most common fragmentation technique for synthetic peptides or enzymatically generated protein fragments, its applicability to analyse peptides containing post-translational modifications is in some cases problematic[42]. The superiority of radical-driven fragmentation techniques such as electron-capture dissociation or electron-transfer dissociation (ETD) has recently been established for CID-labile modifications such as glycosylation and phosphorylation. The literature reports that these methods have led to successful site-assignments for phosphorylation (pArg and pLys) and glycosylation (N-glycosylation and O-glycosylation) events[16,43–47]. CID was recently used in an analysis of pCys in two potential transcriptional regulators, SarA and MgrA, whose sequences are thought to harbour phosphorylated Cys residues[26]. Although CID fragmentation showed insufficient sequence coverage with mainly unphosphorylated fragment ions due to extensive neutral losses, small intense intact fragments allowed to assign the phosphate moiety to the Cys residue[26].

This led to our first attempts to fragment peptide **6a** using higher-energy collisional dissociation (HCD), a beam-type CID fragmentation method. Unfortunately, complete neutral loss of phosphate was observed and only unphosphorylated b- and y-type fragments were detected (Supplementary Fig. 19). In contrast, when using ETD with supplemental activation (SA), that is, EThcD, we detected some c and z-type phosphorylated fragment ions, which allow us to confirm the phosphorylation at Cys (Supplementary Fig. 20). The supplemental energy used in EThcD improves the fragmentation of the charge-reduced precursor ion[48]. Next, we synthesized a naturally occurring pCys peptide, a tryptic pCys fragment peptide from the MgrA regulator, which had previously been identified through CID (EQLpCFSLYNAQR)[26]. This peptide (**6b**) was synthesized following the above mentioned protocol, using phosphite **3d** and final ultraviolet light exposure to deliver **6b** in 5% overall yield including SPPS after purification by semi-preparative HPLC (Supplementary Fig. 21). The fragmentation of peptide **6b** using EThcD yielded fragment ions covering the complete peptide sequence (Fig. 3d). Importantly, the unphosphorylated fragment c3 and the phosphorylated fragment c4 were identified, which unambiguously confirmed the phosphorylation at the Cys residue. None of the fragments showed any indication of a transfer to other potential phosphoacceptors.

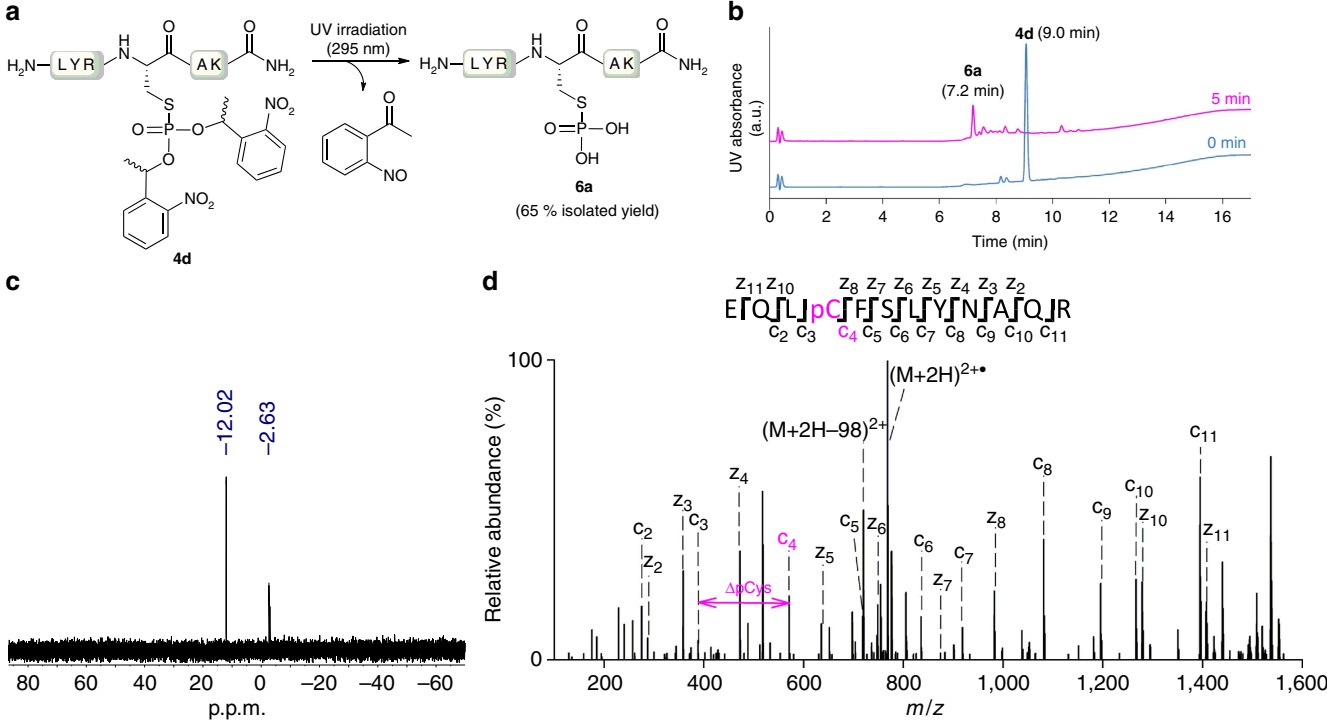

**Figure 3 | Ultraviolet-deprotection and analytical characterization of pCys peptide 6a and 6b.** (**a**) Light-induced photolysis of peptide **4d** to deliver pCys peptide **6a** in 65% isolated yield. (**b**) UPLC-UV traces before (in blue) and directly after ultraviolet light exposure (in magenta). (**c**) $^{31}$P-NMR experiment showing the characteristic peak for pCys residues (12.02 p.p.m.) and inorganic phosphate. (**d**) EThcD MS/MS spectra of pCys peptide **6b**, showing complete sequence coverage and diagnostic fragment ions c4, and z10. a.u., arbitrary unit.

**MS characterization of an endogenous pCys peptide**. Following our success at using EThcD to characterize a synthetic Cys-phosphorylated peptide, we applied the method to identify a pCys in a real biological sample. We chose the cytosolic IIB$^{Glc}$ domain of the transmembrane subunit of the glucose specific transporter IICB$^{Glc}$ from *Escherichia coli* (*E. coli*). IICB$^{Glc}$ is a member of the PEP-dependent PTS and mediates the uptake into *E. coli* and concomitant phosphorylation of glucose[49]. Several studies have shown that glucose is phosphorylated after the phosphoryl moiety has been transferred from PEP through a cascade involving three pHis intermediates (EI, HPr, IIA$^{Glc}$) and IIB$^{Glc}$, which is a pCys protein intermediate (Fig. 4a). Evidence of the pCys IICB$^{Glc}$ intermediate has been demonstrated using NMR, isotopic labelling strategies and stability studies[21,22,24]. CID-MS/MS has also been used to acquire fragment ion spectra after enzymatic digestion of IIB$^{Glc}$ but an unambiguous assignment of phosphorylation to the Cys side-chain has been difficult because these experiments mainly generated unphosphorylated *y*-type fragments with almost complete 80 Da neutral losses[23].

We first overexpressed the four subunits required for the phosphorylation event (EI, HPr, IIA$^{Glc}$ and IIB$^{Glc}$) from the PTS. Without purifying the protein subunits any further, we induced *in vitro* phosphorylation by adding an excess of PEP to a cell lysate containing the four subunits in addition to 10 mM PBS buffer (pH 7.5), sodium orthovanadate, dithiothreitol and MgCl$_2$. Afterwards, we applied a bottom-up proteomic approach using SDS–polyacrylamide gel electrophoresis (PAGE) protein separation in combination with in-gel tryptic digestion (Fig. 4b), or in-solution tryptic-digestion followed by a chromatographic separation of peptides (Fig. 4c). The samples produced by both methods were analysed through nLC-ESI-EThcD MS/MS (Fig. 4b,c). The pCys tryptic peptide ENITNLDApCITR was identified, together with the corresponding unphosphorylated

peptide (Fig. 5a,b and Supplementary Fig. 22). The MS/MS spectra showed complete sequence coverage of the peptide, permitting an unambiguous localization of the phosphorylation site to the Cys residue (Fig. 5c). We observed no neutral losses or phosphate rearrangements, another indication of the high-stability of the phosphorothiolate bond in the gas-phase during the EThcD process. The synthetic version (**6c**) of the endogenous peptide was synthesized (Supplementary Fig. 23) and fragmented by EThcD showing the same fragmentation pattern as observed for the endogenous ENITNLDApCITR peptide (Supplementary Figs 24 and 25).

## Discussion

We have developed a novel chemoselective phosphorylation strategy that enables us to incorporate phosphorylated Cys residues on unprotected peptides in a stereochemically defined way. This chemical tool allowed us to synthesize pCys peptides that could be used as probes for the establishment of an advanced EThcD method that permits a reliable characterization and assignment of this important, specific type of protein phosphorylation. As a proof of principle, we used the EThcD method in combination with a bottom-up proteomic approach to characterize an endogenous pCys peptide from *E.coli*. These results show that the presented chemical and analytical tools are highly valuable in accessing endogenous pCys peptides and thereby open the possibility to identify new Cys phosphorylation sites from biological samples.

## Methods

**General methods.** All reagents, starting materials, amino acids and solvents were purchased from commercial suppliers and used without further purification if not further mentioned. $^1$H and $^{31}$P-NMR spectra were recorded on a Bruker Ultrashield AV 600 MHz at ambient temperature. The chemical shifts are reported in

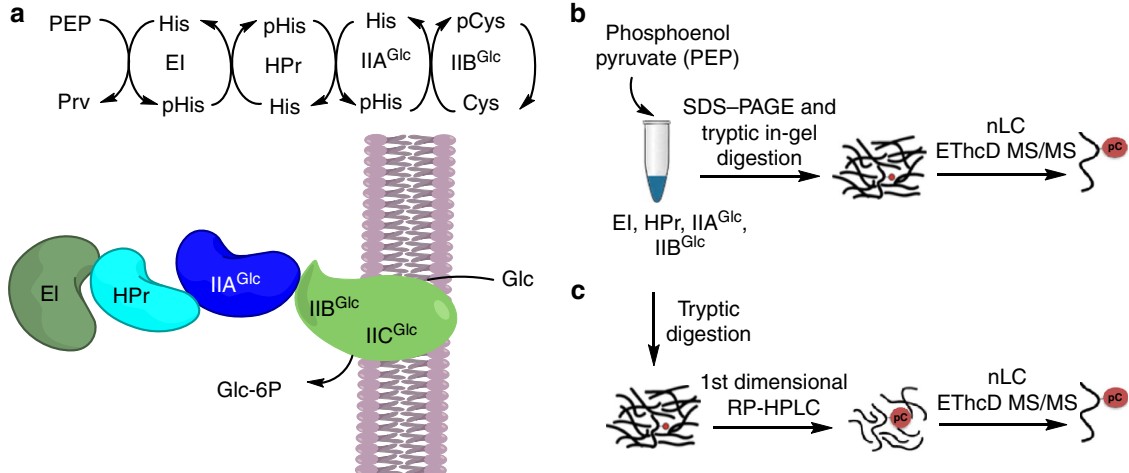

**Figure 4 | Characterization of an endogenous pCys peptide by EThcD MS/MS.** (**a**) Glucose-specific phosphoenolpyruvate-dependent phosphotransferase system. The phosphate group is transfered from PEP sequentially to the IIB<sup>Glc</sup> domain of the membrane protein IICB<sup>Glc</sup>. (**b**) SDS–PAGE in-gel trypsin digeston approach. (**c**) In solution tryptic digestion and offline two-dimensional LC approach.

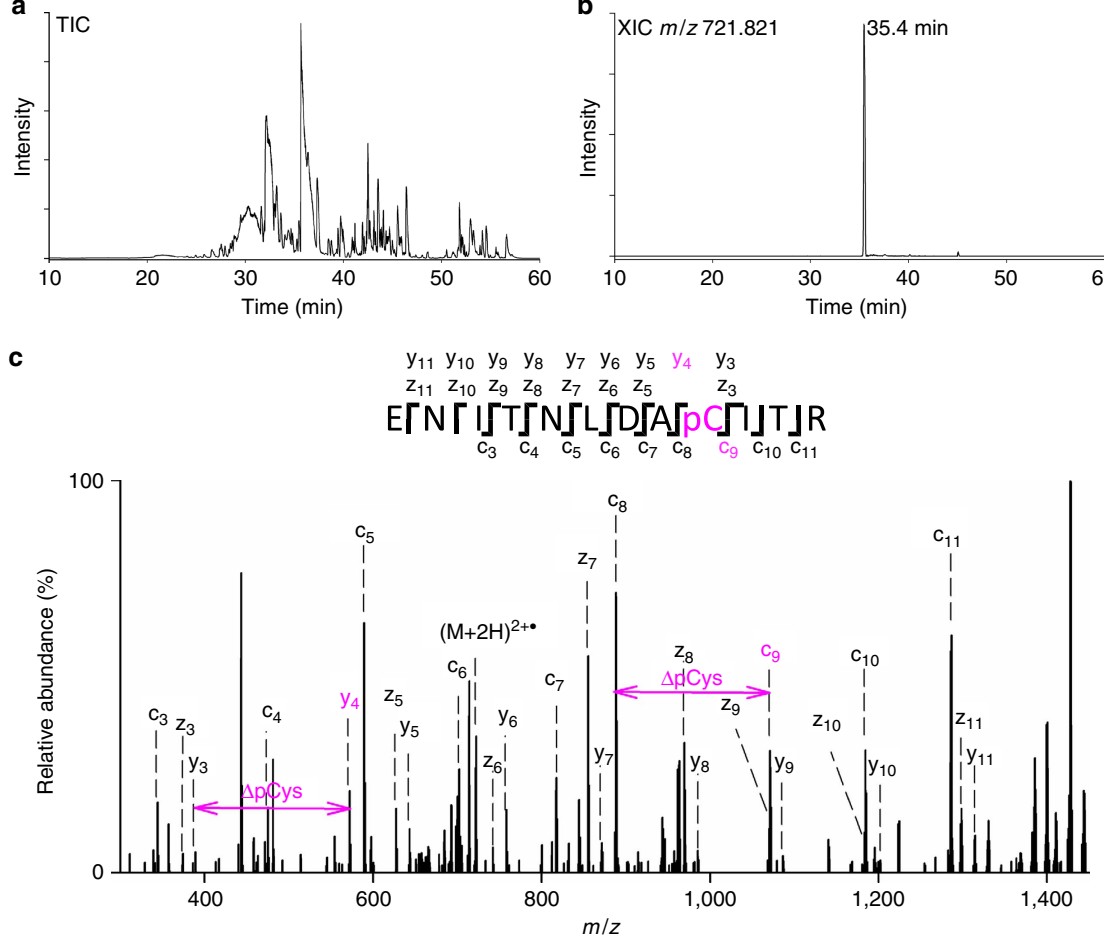

**Figure 5 | Analysis by nLC-ESI-EThcD MS/MS of the endogenous pCys peptide.** (**a**) Total ion chromatogram (TIC) of the tryptic digestion after *in vitro* phosphorylation. (**b**) Extracted ion chromatogram (XIC) of the phosphorylated Cys peptide with $m/z$ 721.821. (**c**) EThcD MS/MS spectra of the endogenous pCys peptide, ENITNLDApCITR, showing complete sequence coverage and diagnostic fragment ions $c_9$, $y_4$ and $z_5$.

p.p.m. relatively to the residual solvent peak. UPLC-UV traces were obtained using a Waters H-class instrument, equipped with a Quaternary Solvent Manager, a Waters autosampler, a Waters Tunable UV detector connected to a 3,100 mass detector using a Acquity UPLC-BEH C18 1.7 μM 2.1 × 50 mm reversed-phase column with a flow rate of 0.6 ml min$^{-1}$. The following solvent and gradients were applied for all peptides if not further mentioned: method 1 (A = H$_2$O + 0.1% TFA,

B = MeCN + 0.1% TFA) 1% B 0–5 min, 1–99% 5–15 min, 99% B 15–17 min. Method 2 (A = H$_2$O + 0.1% TFA, B = MeCN + 0.1% TFA) 1% B 0–3 min, 1–99% 3–8 min, 99% B 8–10 min. UPLC-UV chromatograms were recorded at 220 nm (Supplementary Figs 26–39). Cys-containing, activated Ellman-disulfide and phosphorothiolate ester peptides were purified on a Dionex 580 HPLC system using a reversed phase Nucleodur C18 HTec column (10 × 250 mm) at a flow rate

of 2 ml min$^{-1}$ using a TFA gradient (0% B 0–5 min; 0–50% B 5–60 min; 50–100% 60–80 min; 100% B 80–90 min) in H$_2$O/MeCN system (A = H$_2$O + 0.1% TFA, B = 85% MeCN + 15% A + 0.1% TFA) (Method A). Phosphorylated cysteine peptides were purified on a Shimadzu HPLC system using a reversed phase Nucleodur C18 HTec column (10 × 250 mm) at a flow rate of 3 ml min$^{-1}$ using an MeCN/H$_2$O gradient system (0% B 0–5 min; 0–40% B 5–55 min; 40–100% B 55–85 min) in alkaline aqueous buffer (pH 8.2) as a mobile phase (A = 10 mM ammonium acetate in H$_2$O, B = MeCN + 10% of sol. A) (Method B). Characterization of peptides were done with an Agilent 6,210 ToF liquid chromatography (LC)/MS system and with an Orbitrap Fusion mass spectrometer (Thermo Scientific) (Supplementary Figs 40–55).

**SPPS.** For the synthesis of all peptides in this study, standard Fmoc (9-fluorenyl-methoxycarbonyl)-based SPPS was used on Rink Amide or Wang resins. Peptides were synthesized either manually or with a Tribute Peptide Synthesizer (Protein Technologies, Inc) via standard Fmoc-based conditions with HOBt/HBTU/DIPEA activation and piperidine Fmoc deprotection in DMF. Peptides were cleaved and isolated by reversed-phase-HPLC and verified by electrospray ionization mass spectroscopy (Supplementary Figs 40–43). For the specific procedures used for the synthesis of the Cys-containing peptides **1a–e**, see the Supplementary Methods.

**Phosphite synthesis.** Synthesis of phosphite esters **3b–3e** has been described previously[9,10].

**Synthesis of phosphorothiolate ester peptides.** A stock solution of peptide **2a** (2.5 mM) in DMF or in 25 mM Tris-HCl buffer (pH 7.2) was prepared. All phosphites employed (**3a–e**) were added in excess (5 eq.). Reactions in DMF and Tris buffer were run until peptide **2a** was fully or almost depleted (∼16 h in DMF and 3 h in Tris-HCl buffer) at room temperature. After the removing of solvents, the crude reaction mixtures were diluted in a mixture of H$_2$O/MeCN (8:2) and injected in a UPLC-UV and -MS detector. Identity of the product was confirmed by high-resolution-MS and NMR (Supplementary Figs 48–52 and 56–59). For the specific procedures used for the synthesis of the phosphorothiolate ester peptides **4** and **5**, see the Supplementary Methods.

**Epimerization studies.** A stock solution (2.5 mM) for each peptide tripeptide (**2b** or **2c**) in DMF or in 25 mM Tris-HCl buffer (pH 7.2) was prepared. Phosphites employed (**3a, 3b, 3d**) were added in excess (5 eq.). Reactions in DMF and Tris buffer were run until peptide **2b** or **2c** was fully or almost depleted (∼16 h in DMF and 3 h in Tris buffer) at room temperature. After the removing of solvents, the crude reaction mixtures were diluted in a mixture of H$_2$O/MeCN (8:2) and injected in a UPLC-UV and -MS detector. Retention times for each diastereoisomer phosphorothiolate ester tripeptide were checked and compared (**4f** versus **4g**, **4h** versus **4i**, and **4j** versus **4k**). Analytical co-injections by UPLC-UV (Method 1) and -MS were performed for each pair of diastereoisomers. Identity of the products was confirmed by MS.

**Ultraviolet-irradiation of *o*-nitrobenzyl-based phosphorothiolate ester peptides.** Tris-HCl buffered (20 mM, pH 7.8) solution (500 μl) of peptide **4d** (1.8 mM), **4l** (0.96 μM), and **4m** (0.95 μM) were irradiated at 295 nm with a ultraviolet lamp for 5 min and **4e** (1.4 mM) for 15 min. The samples were injected into the UPLC-UV and the identification of the peaks was confirmed by high-resolution-MS and HCD fragmentation (Supplementary Figs 53–55 and 60).

**Stability studies of phosphocysteine peptides.** A solution stock of substrate **6a** (0.45 μM) in 10 mM Tris-HCl buffer (pH 7.5) was prepared. Inosine (1 mM) was added as standard to monitor the decay of substrate **6a** under different pH. Aliquots (6 μl) were taken every 2 h and analysed by UPLC-UV (220 nm).

**Tandem MS (MS/MS) analysis.** For LC–MS analysis, peptides were dissolved in water (1 pM μl$^{-1}$) and analysed by a reversed-phase capillary liquid chromatography system (Dionex Ultimate 3,000 NCS-3500RS Nano, Thermo Scientific) connected to an Orbitrap Fusion mass spectrometer (Thermo Scientific). LC separations were performed on an in-house packed 75 μm inner diameter PicoTip column containing 25 cm of ReproSil-Pur C18AQ resin (3 μm, 120 Å, Dr Maisch GmbH Ammerbuch-Enttringen, Germany) at an eluent flow rate of 300 nl min$^{-1}$ using a gradient of 2–50% B in 40 min. Mobile phase A contained 0.1% FA in water, and mobile phase B contained 0.1% FA in MeCN. Fourier transformed (FT) survey scans were acquired in a range from 350 to 1,500 *m/z*, with a resolution of 60,000, at an automatic gain control target of 100,000 and a max injection time of 50 ms. In data-dependent mode monoisotopic precursor ions with charge states between 2 and 5 were selected, preferential choosing higher charge states before fragmentation. FTMS2 spectra were measured in the orbitrap with a resolution of 15,000, an automatic gain control target of 100,000 and a max injection time of 200 ms. For HCD the normalized collision energy was set to 30%. EThcD spectra were collected using calibrated charge-dependent ETD parameters and HCD SA was enabled. For EThcD fragmentation the SA collision energy was set to 10%, 20% or 30%, respectively. MS/MS spectra were manually verified and compared with the

theoretical fragment ions of the peptides considering all possible phosphorylation sites.

**Overexpression of EI, HPr, IIA$^{Glc}$ and IIB$^{Glc}$.** The EI construct was expressed from the plasmid PMS EH2-EI in WA2127ΔHIC *E. coli* cells. Cells from glycerol-stock, stored at − 70 °C, were picked and grown overnight in lysogeny broth (LB) medium supplemented with ampicilin (100 μg ml$^{-1}$). One litre of freshly prepared LB-ampicilin medium was then inoculated with 10 ml overnight culture. The cells were grown at 37 °C in an orbital shaker to an OD550 of ∼0.8, induced with isopropyl-β-D-thiogalactoside (IPTG; (100 μM final concentration) and then allowed to grow for another 5–6 h at 37 °C. Cells from 2 l. LB-media were then harvested by centrifugation (7,000g, 30 min, 4 °C) and stored at − 20 °C.

The HPr construct was expressed from the plasmid PtsH9 in WA2127ΔHIC *E. coli* cells. Cells from glycerol-stock, stored at − 70 °C, were picked and grown overnight in LB medium supplemented with ampicilin (50 μg ml$^{-1}$). One litre of freshly prepared LB-ampicilin medium was then inoculated with 10 ml overnight culture. The cells were grown at 37 °C in an orbital shaker to an OD550 of ∼0.8, induced with IPTG (100 μM final concentration) and then allowed to grow for another 5–6 h at 37 °C. Cells from 2 l. LB-media were then harvested by centrifugation (7,000g, 30 min, 4 °C) and stored at − 20 °C.

The IIA$^{Glc}$ construct was expressed from the plasmid Crr-pMS470 D8 in WA2127ΔHIC *E. coli* cells. Cells from glycerol-stock, stored at − 70 °C, were picked and grown overnight in LB medium supplemented with ampicilin (50 μg ml$^{-1}$). One litre of freshly prepared LB-ampicilin medium was then inoculated with 10 ml overnight culture. The cells were grown at 37 °C in an orbital shaker to an OD550 of ∼0.8, induced with IPTG (100 μM final concentration) and then allowed to grow for another 5–6 h at 37 °C. Cells from 2 l LB-media were then harvested by centrifugation (7,000g, 30 min, 4 °C) and stored at − 20 °C.

The IIB$^{Glc}$ construct was expressed from the plasmid pet28H6 IIBGlc in BW25113ΔacrB *E. coli* cells. Cells from glycerol-stock, stored at − 70 °C, were picked and grown overnight in LB medium supplemented with kanamycine (50 μg ml$^{-1}$). One litre of freshly prepared LB-kanamycine medium was then inoculated with 10 ml overnight culture. The cells were grown at 37 °C in an orbital shaker to an OD550 of ∼0.8, induced with IPTG (100 μM final concentration) and then allowed to grow for another 3.5 h at 37 °C. Cells from 2 l. LB-media were then harvested by centrifugation (7,000g, 30 min, 4 °C) and stored at − 20 °C.

**Cell lysate preparation.** In all, 0.6 g for each cell pellet (2.4 g) was mixed together and resuspended in 100 ml 10 mM PBS solution containing 1 mM dithiothreitol, 1 mM Na$_3$VO$_4$ and one complete EDTA-free protease inhibitor tablet (Roche Diagnostics GmbH). The cells were lysed via sonication for 6 min. After ultra-centrifugation (18,000g for 25 min, 4 °C) the supernatant was collected, was filtered and aliquots were frozen in liquid nitrogen and stored at − 80 °C until further use (48 mg ml$^{-1}$ total protein concentration measured by Nanodrop).

***In vitro* phosphorylation.** A volume of 150 μl of 20 mM PBS buffer containing MgCl$_2$ (1.3 mM), dithiothreitol (5 mM) and PEP (20.3 mM) were added to 50 μl of the cell lysate (see section 'cell lysate preparation', final total protein concentration was 12 mg ml$^{-1}$ (Nanodrop)). After 1 h incubation at room temperature, a sample (2.5 μl) of the mixture was either submitted for SDS–PAGE and further in-gel trypsin digestion approach or alternatively used as a standard for in solution trypsin digestion.

**Proteomic identification of an endogenous pCys peptide by an in-gel tryptic digestion approach and LC–MS/MS.** Following *in vitro* phosphorylation with PEP, the cell lysate was analysed by SDS–PAGE gel electrophoresis. The gel band at the expected shift (10 kDa) was cutted and a standard in-gel tryptic digestion approach was performed[50]. Samples were re-suspended in 1% MeCN aqueous solution and were submitted to nanoLC–MS/MS analysis using a C18 stationary phase. Amino-acid sequence was confirmed by HCD fragmentation (Supplementary Fig. 61) (see Tandem MS analysis section).

**Proteomic identification of an endogenous pCys peptide by an offline two-dimensional LC.** Following *in vitro* phosphorylation with PEP, the cell lysate was digested with 20 μg of trypsin (sequencing grade, Promega) overnight. The digested cell lysate was then submitted to a LC separation on a Acclaim Pep Map 100 C18 column (1.0 mm × 25 cm × 5 μm) at an eluent flow rate of 40 μl min$^{-1}$ using a linear gradient of 1–50% B in 80 min. Mobile phase A contained 0.1% TFA and 5% MeCN in water, and mobile phase B contained 0.1% TFA and 20% water in MeCN. Eluent was collected each 65 s and combined to 16 fractions. Fractions from sequential elution section were pooled (for example, 1 + 17, 2 + 18, 3 + 19, …). The samples were dried under vacuum, reconstituted in 6 μl of 1% MeCN, and submitted to nanoLC–MS/MS analysis using a C18 stationary phase (see Tandem MS analysis section).

**Supplementary methods.** Synthetic protocols and characterization of peptides, UPLC-UV chromatogram traces, EThcD, HCD and NMR spectra are described and shown in the Supplementary Information.

**Data availability.** The data that support the findings of this study are available from the corresponding author upon reasonable request.

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

## Acknowledgements

We thank H. Stephanowitz for assistance with the offline two-dimensional LC separation, M. Schümann for assistance with mass spectrometry, Anett Hauser for assistance with phosphite synthesis, and B. Erni for the donation of the plasmids and helpful discussions. We acknowledge support from the DFG (SFB 765 and SPP 1623), the Fonds der Chemischen Industrie, the Einstein Foundation and the Boehringer-Ingelheim Foundation (Plus 3 award).

## Author contributions

C.P.R.H. and J.B-V. conceived and designed the research. J.B-V. synthesized peptides, performed the quantification studies and the *in vitro* phosphorylation, and designed and executed the MS-based proteomics experiments, together with E.K. M.P. performed the nLC MS/MS and the EThcD experiments, as well as analysed the data obtained. O.N-G.

performed the stability studies. P.S. performed the $^{31}$P NMR and $^1$H, $^{31}$P HMBC NMR experiments. J-M.J. provided the plasmids for EI, HPr, IIA$^{Glc}$ and IIB$^{Glc}$ proteins. J.B-V., E.K. and C.P.R.H. co-wrote the manuscript, with comments and contributions from all authors.

## Additional information

**Competing financial interests:** The authors declare no competing financial interests.

