## [Peer Review File · Nature Communications]

Reviewers' comments:

Reviewer #1 (Remarks to the Author):

This report by Hackenberger and coworkers makes a significant contribution to the study of atypical (non O-linked) phosphorylation events in proteins. The authors describe a new strategy for the generation of enantiomerically-pure phosphocysteine residues. This strategy enables the authors to generate sufficient material for development of mass spectrometric methods for ID of the modified residues in proteins suitable for application in proteomics workflows. Finally, they use this approach to confirm the presence of a phosphocysteine residue in a known phosphocysteine-containing protein.

The work is original, no defined methods for the generation of phosphocysteine have been reported previously other than the dHA-mediated route described by Davis and co-workers. The MS methods are also a valuable and original contribution. Given this I am happy to recommend the paper for acceptance with little, if any modification other than those detailed below.

The data, methodology and presentation are all excellent - detailed experimental is provided and this is of an appropriate standard in the main text. As detailed below, the manner in which the supplementary information is presented is confusing.

There are no issues with the conclusions of the paper other than the caveats given at the end of this review.

I would suggest that the way in which supplementary figures and experimental is presented is inappropriate. In reading the main text I was surprised to encounter supplementary figure numbers in the 50s - this is due to embedding of these figures within a chemical experimental in a linear fashion. If figures are sufficiently important to merit inclusion in the discussion then these should be fully formatted figures of a quality appropriate with inclusion in the main body of the text rather than single graphs/HPLC traces. These figures should be numbered separately and formatted as a separate section of the supplementary information so that they can be found quickly! Other figures should be numbered sequentially after this. (NB Is it really necessary for every HPLC trace to have a figure number!)

Appropriate references are used throughout.

Finally, the authors refer to their phosphorylation chemistry as being 'site-selective'. This is true, in so much that it is chemoselective as that other residues (Ser, Arg, Lys, Tyr) are not modified by the chemistry. However it is not truly site-selective - I would interpret this as meaning that a particular cysteine residue was modified in the presence of other unprotected cysteine residues. This is not the case and the language in the paper needs to be modified to reflect this.

Reviewer #2 (Remarks to the Author):

Bertran-Vicente (2016) Chemoselective synthesis and analysis of naturally occurring phosphorylated cysteine peptides.

The manuscript by Bertran-Vicente describes the synthesis of several peptides in which the cysteine residue is phosphorylated. They also use the purified peptides to develop an MS based technique (electron transfer higher-energy dissociation (ETHcD)) that they then use to detect a phosphorylated cysteine residue in a known protein. Overall, the manuscript is well written and the conclusions supported by the data, however, the synthetic yields are not quantitative and no new biology is discovered. As such, it is difficult to recommend this manuscript for publication in Nature Comm. It is more suitable for Org Lett or ChemBioChem.

Reviewer #3 (Remarks to the Author):

A. Summary of Key results:

In this article authors developed a chemical method to make phosphorylated cysteine, and used this method to develop a mass spectrometry MS/MS protocol to detect this type of posttranslational modification in vitro and in cells. This is a novel study, since it provides access to detect and possibly quantify this labile PTM.

B. The research is original, and the synthesis of pCys peptides is novel, and provides advantage over the existing dehydroalanine chemistry that loses peptide chirality.

C. All good.

D. All Good.

E. See "F".

F. Overall this is an excellent manuscript but I have two minor points.

a) page 8 the authors conclude that YCA (1b, L-Cys; 1c, D-Cys) reacts with the phosphine and this reaction is epimerization free. They conclude so because reaction products 4h and 4i have different retention time. While intuitively correct, this conclusion does not rule out DHA pathway.

First the authors should clarify what is the mechanism that would lead to DHA under given conditions. Second, 1b and 1c could still undergo elimination to produce DHA, followed by addition of thiophosphate, and this addition could be diastereoselective for whatever reason. In this regard preparing DHA of 1b and 1c and conducting the addition reaction with thiophosphate (or the corresponding thiophosphate ester) would be useful to show that indeed DHA of this peptide does undergo addition reaction and it is not diastereoselective.

Subsequently authors show that they can detect endogenous ENITNLDApCITR peptide. (Fig. 5) Since this is a new method, I would recommend to be 100% sure to make synthetic ENITNLDApCITR peptide and show in vitro that the MS/MS fragmentation pattern is the same as observed for the endogenous peptide.

I would also recommend to make heavy ENITNLDApCITR add it to the digested cell lysate followed by MS/MS fragmentation to make sure that cell lysate does not affect ENITNLDApCITR fragmentation pattern. However I leave this experiment at the discretion of the authors.

Overall this is an excellent manuscript and I recommend its publication in Nature Communications.

G. Appropriate.

H. All good.

Answers to the reviewers:

Reviewer 1

Comments:

This report by Hackenberger and coworkers makes a significant contribution to the study of atypical (non O-linked) phosphorylation events in proteins. The authors describe a new strategy for the generation of enantiomerically-pure phosphocysteine residues. This strategy enables the authors to generate sufficient material for development of mass spectrometric methods for ID of the modified residues in proteins suitable for application in proteomics workflows. Finally, they use this approach to confirm the presence of a phosphocysteine residue in a known phosphocysteine-containing protein.

The work is original, no defined methods for the generation of phosphocysteine have been reported previously other than the dHA-mediated route described by Davis and co-workers. The MS methods are also a valuable and original contribution. Given this I am happy to recommend the paper for acceptance with little, if any modification other than those detailed below.

The data, methodology and presentation are all excellent - detailed experimental is provided and this is of an appropriate standard in the main text. As detailed below, the manner in which the supplementary information is presented is confusing.

There are no issues with the conclusions of the paper other than the caveats given at the end of this review.

I would suggest that the way in which supplementary figures and experimental is presented is inappropriate. In reading the main text I was surprised to encounter supplementary figure numbers in the 50s - this is due to embedding of these figures within a chemical experimental in a linear fashion. If figures are sufficiently important to merit inclusion in the discussion then these should be fully formatted figures of a quality appropriate with inclusion in the main body of the text rather than single graphs/HPLC traces. These figures should be numbered separately and formatted as a separate section of the supplementary information so that they can be found quickly! Other figures should be numbered sequentially after this. (NB Is it really necessary for every HPLC trace to have a figure number!)

Appropriate references are used throughout.

Finally, the authors refer to their phosphorylation chemistry as being 'site-selective'. This is true, in so much that it is chemoselective as that other residues (Ser, Arg, Lys, Tyr) are not modified by the chemistry. However it is not truly site-selective - I would interpret this as meaning that a particular cysteine residue was modified in the presence of other unprotected cysteine residues. This is not the case and the language in the paper needs to be modified to reflect this.

Response:

We thank the reviewer for recommending publication in *Nature Communications* and for making several useful suggestions to improve the manuscript. We addressed the minor points mentioned by the reviewer and improved the Supporting Information file according to the Nature Communications guidelines. In addition we addressed the use of the "site-selective" terminology in the manuscript (details are given in response to issue 2).

Issues mentioned by the Reviewer 1:

1. I would suggest that the way in which supplementary figures and experimental is presented is inappropriate. In reading the main text I was surprised to encounter supplementary figure numbers in the 50s - this is due to embedding of these figures within a chemical experimental in a linear fashion. If figures are sufficiently important to merit inclusion in the discussion then these should be fully formatted figures of a quality appropriate with inclusion in the main body of the text rather than single graphs/HPLC traces. These figures should be numbered separately and formatted as a separate section of the supplementary information so that they can be found quickly! Other figures should be numbered sequentially after this. (NB Is it really necessary for every HPLC trace to have a figure number!)

Response:

As suggested we have made the corrections on the Supporting Information file changing the formatting as suggested by reviewer and accordingly to *Nature Communications* guidelines.

2. Finally, the authors refer to their phosphorylation chemistry as being 'site-selective'. This is true, in so much that it is chemoselective as that other residues (Ser, Arg, Lys, Tyr) are not modified by the chemistry. However it is not truly site-selective - I would interpret this as meaning that a particular cysteine residue was modified in the presence of other unprotected cysteine residues. This is not the case and the language in the paper needs to be modified to reflect this.

Response:

We thank the reviewer for that indication and agree with the statement that we cannot differentiate several Cys residues in an unprotected peptide using our methodology. Consequently, we have modified the terminology in the following sentences (highlighted in yellow).

In the Abstract section on page 1 “site-selective” was removed and “stereochemically defined” was added:
“We achieve the **stereochemically defined** phosphorylation of a Cys residue”

In the legend of Figure 1b on page 3:

“Site-selective **addition of phosphite to an activated Ellman-disulfide to deliver a stereochemically defined pCys residue**”

In the subheading of Results heading on page 5 “site selective” was removed:

“*Synthesis of pCys peptides*”

In the legend of Figure 2 on page 6 “site selective” was removed and “Chemoselective” was added:

“Figure 2. **Chemoselective** synthesis of phosphorothiolate esters peptides 4 and 5”

In the Conclusion section on page 16 “site selective” was removed and “unprotected” was added:

“We have developed a novel chemoselective phosphorylation strategy that enables us to incorporate phosphorylated Cys residues on **unprotected** peptides in a stereochemically defined way”

Reviewer 2

Comments:

Bertran-Vicente (2016) Chemoselective synthesis and analysis of naturally occurring phosphorylated cysteine peptides. The manuscript by Bertran-Vicente describes the synthesis of several peptides in which the cysteine residue is phosphorylated. They also use the purified peptides to develop an MS based technique (electron transfer higher-energy dissociation (EThcD)) that they then use to detect a phosphorylated cysteine residue in a known protein. Overall, the manuscript is well written and the conclusions supported by the data, however, the synthetic yields are not quantitative and no new biology is discovered. As such, it is difficult to recommend this manuscript for publication in Nature Comm. It is more suitable for Org Lett or ChemBioChem.

Response:

We thank the reviewer for reviewing the submitted manuscript. We respect the opinion of the reviewer about the work here presented, but we strongly disagree that this work is more suitable for the other journals mentioned. In particular we disagree with the statement that the synthetic yields are not quantitative. For instance, reaction of phosphite **3d** with unprotected Ellman-disulfide peptide **2a** in dimethylformamide furnished the corresponding phosphorothiolate ester peptide **4d** in almost quantitative conversion (97% based on MS) and 55% isolated yield (Table 1, entry 4). This is an excellent result for a peptide synthesis protocol after chromatographic purification. Moreover, our work presents a new MS-based method for analyzing pCys peptides, which we foresee will have a tremendous impact on the discovery of new pCys sites and therefore in new findings of pCys biology as supported by referees 1 and 3.

Reviewer: 3

Comments:

A. Summary of Key results:

In this article authors developed a chemical method to make phosphorylated cysteine, and used this method to develop a mass spectrometry MS/MS protocol to detect this type of posttranslational modification in vitro and in cells. This is a novel study, since it provides access to detect and possibly quantify this labile PTM.

B. The research is original, and the synthesis of pCys peptides is novel, and provides advantage over the existing dehydroalanine chemistry that loses peptide chirality.

C. All good.

D. All Good.

E. See "F".

F. Overall this is an excellent manuscript but I have two minor points.

a) page 8 the authors conclude that YCA (1b, L-Cys; 1c, D-Cys) reacts with the phosphine and this reaction is epimerization free. They conclude so because reaction products 4h and 4i have different retention time. While intuitively correct, this conclusion does not rule out DHA pathway.

First the authors should clarify what is the mechanism that would lead to DHA under given conditions. Second, 1b and 1c could still undergo elimination to produce DHA, followed by addition of thiophosphate, and this addition could be diastereoselective for whatever reason. In this regard preparing DHA of 1b and 1c and conducting the addition reaction with thiophosphate (or the corresponding thiophosphate ester) would be useful to show that indeed DHA of this peptide does undergo addition reaction and it is not diastereoselective.

Subsequently authors show that they can detect endogenous ENITNLDapCITR peptide. (Fig. 5) Since this is a new method, I would recommend to be 100% sure to make synthetic ENITNLDapCITR peptide and show in vitro that the MS/MS fragmentation pattern is the same as observed for the endogenous peptide.

I would also recommend to make heavy ENITNLDapCITR add it to the digested cell lysate followed by MS/MS fragmentation to make sure that cell lysate does not affect ENITNLDapCITR fragmentation pattern. However I leave this experiment at the discretion of the authors.

Overall this is an excellent manuscript and I recommend its publication in *Nature Communications*.

G. Appropriate.

H. All good.

Response:

We thank the Reviewer for recommending publication in *Nature Communications* and for making several useful suggestions to improve the manuscript. We addressed the minor points of the reviewer as follows:

Issues mentioned by the Reviewer 3:

1. Page 8 the authors conclude that YCA (1b, L-Cys; 1c, D-Cys) reacts with the phosphine and this reaction is epimerization free. They conclude so because reaction products 4h and 4i have different retention time. While intuitively correct, this conclusion does not rule out DHA pathway. First the authors should clarify what is the mechanism that would lead to DHA under given conditions. Second, 1b and 1c could still undergo elimination to produce DHA, followed by addition of thiophosphate, and this addition could be diastereoselective for whatever reason. In

this regard preparing DHA of **1b** and **1c** and conducting the addition reaction with thiophosphate (or the corresponding thiophosphate ester) would be useful to show that indeed DHA of this peptide does undergo addition reaction and it is not diastereoselective.

Response:

We thank reviewer for raising this interesting point and apologize if our first presentation was not clear. Nevertheless, we think that with the presented data we can indeed rule out the intermediate formation of dehydroalanine (Dha).

First, we would like to clarify the mechanism that would lead to the Dha pathway as requested by the reviewer. Similarly to using phosphines (see Ref. 35), we could envision the elimination of thiophosphate ester from the thiophosphonium cation intermediate, which would finally form the Dha substrate. Subsequent re-addition of the eliminated thiophosphate ester, as also suggested by the reviewer, would indeed deliver the corresponding phosphorothiolate ester peptides (please see below Fig. A to illustrate the **hypothetical** formation of Dha and a subsequent non-diastereoselective re-addition of the thiophosphate ester from peptides **2b** and **2c** reported in the manuscript). Assuming that we would have a **diastereoselective** addition to Dha, as indicated by the reviewer, we would obtain the **same phosphorothiolate ester peptide** when starting independently from two Cys-peptides, which only differ in the stereochemistry at Cys, since the formation of Dha would make the two peptides identical (please see in Fig. A the green arrow). As visible in our experiments, we have indeed performed this experiment and employed a L-Cys tripeptide **1b** and a D-Cys tripeptide **1c**. In our experiments we clearly **observed two different phosphorothiolate ester peptides, 4h and 4i**, (please see Supplementary Figs 5-7) and thereby can rule out the possibility that we have a diastereoselective addition. In the case the re-addition of the eliminated thiophosphate ester would proceed in a non-diastereoselective manner, we would observe a mixture of both phosphorothiolate ester peptides (**4h** and **4i**) for each experiment using L-Cys tripeptide **1b** and D-Cys tripeptide **1c**. Again, we only observe a single stereoisomer in each of this transformation, which differ in the HPLC-retention time.

In conclusion, we think that our experiments with the model tripeptide demonstrate that our synthetic approach does not proceed via the intermediate formation of Dha.

Figure A. (a) Hypothetical Dha formation and conjugate addition of thiophosphate ester to Dha starting from the L-Cys **2b**. (b) Hypothetical Dha formation and conjugate addition of thiophosphate ester to Dha starting from the D-Cys **2c**.

2. Subsequently authors show that they can detect endogenous ENITNLDApCITR peptide. (Fig. 5) Since this is a new method, i would recommend to be 100% sure to make synthetic ENITNLDApCITR peptide and show in vitro that the MS/MS fragmentation pattern is the same as observed for the endogenous peptide.

Response:

We thank reviewer for this comment. As suggested, the synthetic version of the endogenous ENITNLDApCITR peptide was synthesized following the described protocol (peptide **6c**) (Supplementary Fig. 23) and subjected to EThcD fragmentation (Supplementary Fig. 24). Both the endogenous ENITNLDApCITR peptide and peptide **6c** have identical fragmentation pattern (Supplementary Fig. 25).

Consequently, we have added the corresponding sentence (highlighted in yellow) on page 15:

*The synthetic version (**6c**) of the endogenous peptide was synthesized (Supplementary Fig. 23) and fragmented by EThcD showing the same fragmentation pattern as observed for the endogenous ENITNLDApCITR peptide (Supplementary Figs 24 and 25).*

3. I would also recommend to make heavy ENITNLDApCITR add it to the digested cell lysate followed by MS/MS fragmentation to make sure that cell lysate does not affect ENITNLDApCITR fragmentation pattern. However I leave this experiment at the discretion of the authors.

Response:

We thank the reviewer for the suggestion of this experiment. Nevertheless, we can observe identical fragmentation patterns of the newly synthesized peptide **6c** and the endogenous ENITNLDApCITR peptide. In our opinion, this already demonstrates that the cell lysate does not affect the endogenous pCys peptide and thereby its fragmentation pattern.

REVIEWERS' COMMENTS:

Reviewer #3 (Remarks to the Author):

The authors have addressed all my concerns. Congratulations with the manuscript! I recommend to publish!